# Augmented Reality to Engage Visitors of Science Museums through Interactive Experiences

Elena Spadoni [1], Sara Porro [2], Monica Bordegoni [2], Ilaria Arosio [3], Laura Barbalini [3] and Marina Carulli [2,*]

1 Department of Design, Politecnico di Milano, 20158 Milano, Italy; elena.spadoni@polimi.it
2 Department of Mechanical Engineering, Politecnico di Milano, 20156 Milano, Italy; sara.porro@polimi.it (S.P.); monica.bordegoni@polimi.it (M.B.)
3 INAF-Osservatorio Astronomico di Brera, 20121 Milano, Italy; ilaria.arosio@inaf.it (I.A.); laura.barbalini@inaf.it (L.B.)
* Correspondence: marina.carulli@polimi.it

**Abstract:** In the last years, interactive exhibitions based on digital technologies have become widely common, thanks to their flexibility and effectiveness in engaging visitors and creating memorable experiences. One of the topics in which digital technologies can be particularly effective is the communication of abstract concepts that are difficult for the human mind to imagine. An emblematic example is the astronomy discipline, which requires us to imagine and understand phenomena far away from our everyday life. In this paper, the authors present a research project, MARSS, in which digital technologies are used effectively to enhance the Users' Experience of the Museo Astronomico di Brera located in Milan. Specifically, the MARSS project aims at designing and developing a new digital journey inside the museum to allow different categories of visitors to enjoy the exhibition in an engaging and interactive way. The paper presents the design and development phases of the experience and its evaluation with users. The results of the evaluation indicate that the digital interactive experience is appreciated by users and is successful in translating the content of high scientific value into more engaging and easily understandable elements.

**Keywords:** science museums; cultural heritage; interactive exhibitions; user experience; extended reality; augmented reality

## 1. Introduction

Museums are fundamental cultural institutions of modern society, and their main role consists in the conservation, preservation, and dissemination of knowledge related to human history, figurative art, craft, applied sciences, technology, and many other topics.

In the last decades, museums have been dealing with the innovation of methods and tools for the communication and dissemination of knowledge. Consequently, they have been subjected to important changes in the approach they use to involve visitors. Among the other approaches, storytelling and interactive exhibitions are two of the most used to make the exhibition more interesting for the visitors, giving them the possibility to interact with pieces of art and objects. Nowadays, interactive exhibitions in museums are becoming broadly common, thanks to their flexibility and effectiveness in engaging visitors in the experience. The primary objective of their use is to make the visit of museums much more engaging, immersive, and suitable for different types of visitors. In particular, visitors become active "consumers", able to define their journey and experience at the museum.

With this approach, the target audience can be "extended" to non-expert visitors who can better understand concepts often considered very complex through these new communication methods. In this sense, the possibility of viewing content instead of reading or listening to the explanation, or rather of looking at images/animations instead of reading texts, reduces the need to design for experienced and knowledgeable users and increases the possibilities of learning and remembering information. In the non-expert visitor's

category, it is also possible to include children, who are often fascinated by science, nature, and art but hardly interested in static representations that are very difficult to understand. On this basis, museums increasingly allow young visitors to "play" with exhibits to acquire new knowledge and enjoy pleasant experiences [1].

For all these reasons, interactive exhibitions are becoming more appreciated and effective.

There are several approaches used to make museum exhibitions interactive. One of the most used approaches to create interactive content in an easy way is the use of digital technologies. Specifically, virtual reality and augmented reality technologies can be effectively integrated into the context of a museum exhibition to support both storytelling and interaction. In particular, by means of augmented reality applications, visitors can enjoy interactive experiences in which additional content, such as the history and the meaning of objects, is digitally added to the real artefacts. Indeed, augmented reality is increasingly used for museum exhibitions to add virtual content to the real ones and "increase" the reality. In this way, objects can self-explain their meaning, correlate with other objects, and so on.

For instance, one of the topics in which augmented reality technology can be particularly effective is the communication of abstract concepts that are difficult to imagine for the human mind. An emblematic example is the astronomy discipline, which requires us to imagine and understand phenomena far away from our everyday life. Nevertheless, humans have always been fascinated by the study of the sky and the laws that govern it, and in recent years there has also been a renewed interest in human space exploration. In addition, there are many connections between the work of the astronomers and the cultural life of the city and the country, not only at the present with technological spin-offs, but also in the past for the compilation of maps or the definition of coordinates, time or whether. An excellent opportunity to gain new knowledge in this area is to visit astronomical museums, such as the Museo Astronomico di Brera, which houses scientific instruments used by astronomers in the past at the Osservatorio Astronomico di Brera.

However, despite the cultural importance of the historic scientific instruments hosted in this museum, the current exhibition does not effectively allow visitors to understand their meanings and uses autonomously. To overcome this limit, the purpose of the research work presented in this paper is to design and develop a new digital journey for the Museo Astronomico di Brera, allowing different categories of users to enjoy the exhibition interactively. In particular, the project aims to innovate the cultural proposal of the museum and give new life to the historical collection, illustrating the role that the Osservatorio Astronomico di Brera has played from its foundation to today, with particular emphasis on the evolution of astronomy, technology, and culture.

## 2. Related Works

### 2.1. Interactive and Digital Exhibitions

New exhibitions aim to raise acknowledgment and curiosity among the visitors, especially children and teenagers. Interactive exhibitions are becoming more and more appreciated and effective, involving people through different senses and calling them to action. Typically, in these cases, the users are prompted to perform an action to see the resulting effects on the environment or on the physical or digital artefact itself. This interactive approach is particularly effective for two reasons. First, it engages the users' body, which is a fundamental aspect of improving the learning process according to the well-known psychological theory of embodied cognition. This theory suggests that cognition is heavily grounded on sensory–motor experiences, meaning that better learning is achieved when the body is involved in the process, as compared to when the experience consists of a mere observation or listening [2]. Second, being actively engaged in the educational process, instead of being the passive recipients of notions, allows the users to catch the deep meaning of the subject-to-be-learned, develop logical and semantic links with related topics, and elaborate a personal and critical vision. Technology can provide a considerable boost to this process, enabling the user to experience the exhibition in a multimodal manner.

For instance, the powerful effects on learning of presenting a topic through the integration of multisensory contents, as compared to reading paper books, is already quite evident. Schools are, in fact, implementing new methods to deliver interactive digital lessons able to attract the students' attention and maximise the probability of storing and retrieving information (a sort of technological application of the so successful Montessori educational principles) [3,4]. Similarly, museums and cultural institutions are more and more modifying their role and approach to improve the users' experience of their exhibitions and make learning new knowledge an exciting and pleasant experience.

Virtual reality (VR) and augmented reality (AR) technologies can be effectively used in the context of a museum exhibition to support both storytelling and interaction [5]. People are becoming increasingly familiar with hybrid realities, making it easier to use this technology for public events and cultural heritage purposes. Many are the examples of the use of these technologies for museum exhibitions. Some museums present digital kiosks or touchscreen tables with texts and information as "side elements" of their exhibitions. These systems, classified as "Standalone installations" by [6], have been extensively used since the 1990s, and are still used, even if the provided interaction is quite poor. In other cases, museum exhibitions effectively rely on interactive digital technologies for communicating content to users. For example, the MONA Museum in Hobart [7] effectively uses the "O" device (an iPod Touch running custom software and housed in a specially designed case) and applications on visitors' smartphones to replace wall texts in the museum and allow visitors to interact with pieces of art. Another interesting example is the Cooper Hewitt Pen at the Cooper Hewitt Smithsonian Design Museum [8]. In this case, the "Pen" device is used by visitors to virtually collect objects around the gallery by touching the interactive tables on which they are placed. The collected objects can be explored in further detail after the visit at home or on visitors' mobile devices. At the Casa Batlló in Barcelona, visits are organised using an AR guide, which provides a dynamic and semi-immersive experience based on a narrative journey to discover the original appearance of the space, the Gaudì's childhood and sources of inspiration [9].

In many other cases, museums develop VR and AR experiences for some special exhibitions [6]. Among the others, in 2015, the British Museum's Samsung Digital Discovery Centre (SDDC) held a Virtual Reality Weekend in the Museum's Great Court, offering the visitors the possibility to enjoy VR environments via immersive and non-immersive devices [10]. Visitors explored a virtual reality Bronze Age site, where three-dimensional scans of objects were presented. Additionally, the Natural History Museum in London featured an interactive VR experience in which visitors were able to virtually handle fossils and other pieces from the Museum's collection in a one-on-one interactive experience [11]. "The Modigliani VR: The Ochre Atelier" experience was developed as part of the Modigliani exhibition at the Tate Modern Museum in London [12] and was integrated into the collection made of paintings, sculptures, and drawings. In this experience, visitors could explore the objects in Modigliani's studio, learning more about his creative process and the materials and techniques he used. In 2017, the Kremer Collection launched "The Kremer Museum", where Dutch and Flemish Old Master paintings from the Kremer Collection are accessible exclusively through virtual reality technology [13]. "Rome Reborn" [14] is an international initiative launched in the mid-1990s by the UCLA Cultural Virtual Reality Laboratory to develop 3D digital models illustrating the urban development of ancient Rome from its foundation to the depopulation of the city in the early Middle Ages. Rome Reborn presented a series of products for personal computers and VR headsets aimed at guiding students and the public in virtual tours of the now-vanished ancient city. Again, "The Private World of Rembrandt" is an AR exhibition concerning the personal story of Rembrandt. It is based on original seventeenth-century documents owned by Amsterdam City Archives and original works by Rembrandt. In detail, by using a specifically developed AR application, visitors were able to frame the original documents, visualise AR contents and listen to a voiceover that tells the story behind the document [15]. In 2017, the Franklin Institute in Philadelphia presented an AR application to enjoy the Terracotta Warrior exhibition in a

more interactive way [16]. The AR application allowed visitors to use their smartphones to scan items and visualise digital reproductions of the clay soldiers and their bronze weapons in their original conditions. Similarly, the Holocaust Memorial Museum in Washington D.C. presented in 2018 an AR application to describe the lives of the villagers of Eisiskes, a town with a large Jewish community who were executed by German soldiers in 1941, prior to their executions [17]. Many other examples and case studies can be found in [6,18–20], which present comprehensive surveys and analyses of the state of the art of virtual, augmented, and mixed reality technologies and applications for cultural heritage. In addition, comparative evaluations of virtual reality museum systems are presented in [21,22]. In particular, it is worth reporting that some critical issues regarding the technological aspects of these systems should still be addressed. For example, virtual, augmented, and mixed reality systems should be developed to be flexible and adaptable to different contexts, easily manageable by museum staff, and quickly upgradeable to prevent rapid obsolescence.

The examples above demonstrate the growing interest in the HCI for the cultural heritage field, as demonstrated by the increasing number of publications [23]. Despite this, there is no consensus and quantitative data on the economic impact of their use in museums. However, several research works presented evaluations of the user experience of digital museums [24,25] and the possible effects of virtual, augmented, and mixed reality applications on museum attractiveness [26].

### 2.2. Science and Technology Museums

Created as "chambers of wonders", over the years, science and technology museums have revised their objectives and innovated their contents according to new communication modalities. As the first important step, in 1969, with the Exploratorium Museum in San Francisco, scientific communication moved from the formal approach of traditional museums to a place open to the public for exploring and discovering. Today all the science museums of modern conception (La Cité des sciences et de L'industrie in Paris, The City of Science in Naples, the MUSE in Trento, and the new sections of the "Leonardo da Vinci" Museum in Milan) are characterised by participatory itineraries for the public. Here, the challenge is to involve visitors in complex cultural experiences, stimulating them at emotional and cognitive levels. So, these museums are becoming public spaces to include scientists and non-scientists, the elderly, and young people. This allows science museums to permanently become part of the cultural tourism offer of cities as other "traditional" types of museums.

However, science and technology museums have been relegated to a secondary role for many years. In 1959, Snow [27] denounced the separation between what is perceived and defined as the two cultures: the scientific and humanistic one. This separation impoverishes people who cannot recognise the common and universal process of knowledge acquisition. This idea of two separate cultures arose in the Age of Enlightenment: Aristotle, Galileo, and Pascal were at the same time philosophers and scientists. Nowadays, we are witnessing an increasing interest in scientific disciplines and, so, in science museums. It is a cultural transformation that foresees a recovery of these disciplines, not only as protagonists of progress and well-being, but also as humanity's cultural heritage.

So, today's context represents an important research area for the design of interactive experiences supported by digital technologies. Science and technology museums are no longer relegated to a minor role on the cultural scene, and they are more and more recognised as important cultural institutions for the dissemination of knowledge. In this context, digital technologies can play a fundamental role in innovating the communication modalities of science museums. As already illustrated, these technologies are increasingly used today for dissemination in museums, allowing the visitor's experience to be interactive and engaging. Having positive effects on the experience and calling the visitor "to action", also makes it possible to improve the learning of the conveyed contents both in qualitative and quantitative terms. Focusing on science topics, it has been proven that interactive

AR applications support the learning process in museum experiences [28] as well as in other contexts [29,30], making "the learning material more vivid and memorable" [31]. To effectively improve learning activities, interactive digital applications should be designed and developed to stimulate the users' "intrinsic motivation" [32], as defined by [33]. So, design principles for digital learning, such as those presented by [34], should be used to design engaging interactive learning experiences. Furthermore, compared to the use of "physical" instruments used to experiment and boost the interaction (as is the case of the Exploratorium Museum), digital technologies offer great flexibility, possibilities of customisation according to the user typology, and can be modifiable and updated over time.

## 3. The MARSS Project

The Museo Astronomico di Brera (MusAB) is a museum collection of the Istituto Nazionale di Astrofisica (INAF). The MusAB is set up at the Osservatorio Astronomico di Brera (Osservatorio), founded around 1760, within the Jesuit College of Brera. The Osservatorio is the oldest scientific institution in Milan. Today, it is part of INAF (http://www.inaf.it, accessed on 22 June 2022), the Italian research centre of excellence for the study of astrophysics and cosmology recognised worldwide. The collection is recognised by Regione Lombardia and it is managed by the Osservatorio in collaboration with the University of Milan.

The museum collection consists of scientific instruments used by astronomers dating back to 250 years and displayed in the Gallery of Instruments (see Figure 1), along with an archive full of well-preserved documents and a library with 35,000 ancient books: outstanding evidence of the scientific and cultural activity carried out in this city.

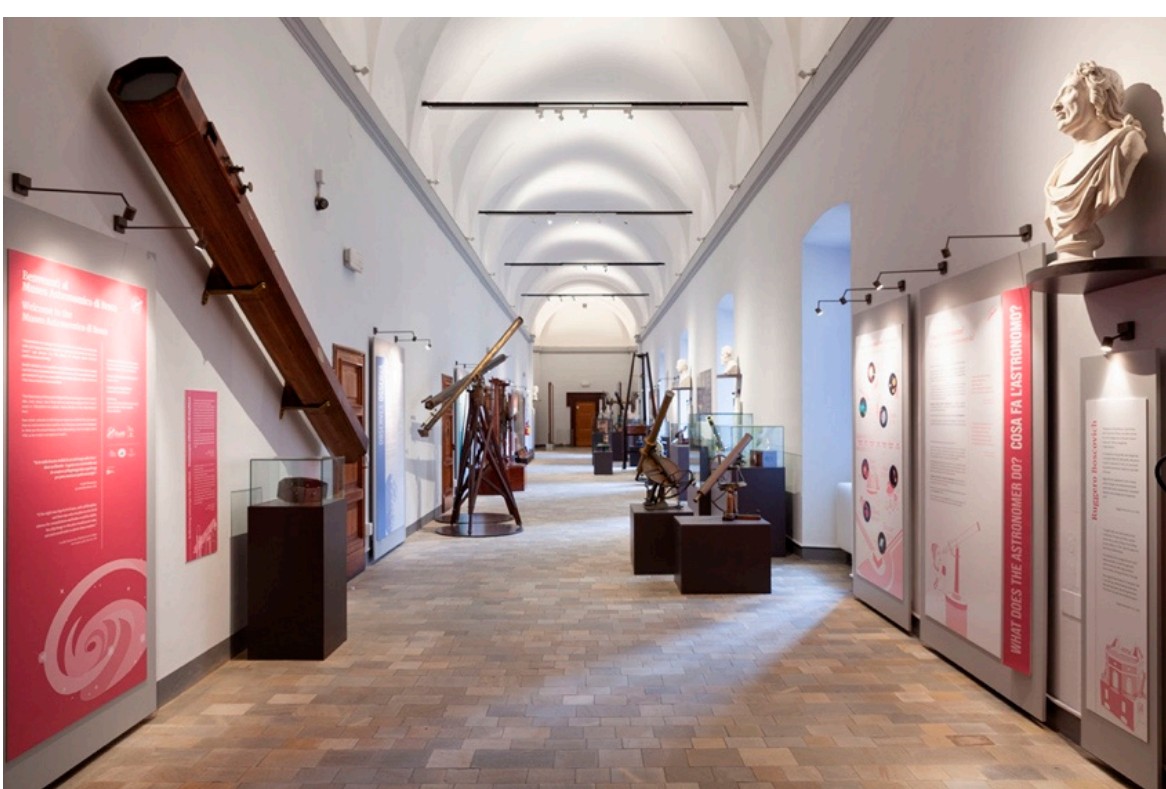

**Figure 1.** The Gallery of Instruments.

Moreover, the famous Cupola Schiaparelli is also part of the MusAB. From this dome, built on top of the Osservatorio, Giovanni Virginio Schiaparelli, former director of the Osservatorio from 1862 to 1900, looked at the Mars planet through the Merz telescope and drew the very first maps of the planet. The dome and the telescope were restored in 1999, and visitors, after the pandemic emergency, are again allowed in during guided tours.

As in many other museums, the exhibition was dated and needed to be improved and modernised. Some improvements have already been made with a new exhibition funded by Regione Lombardia. The new exhibition tries to answer a basic question: "What does the astronomer do?" and then presents, in the different sections, the astronomer's different activities: he/she observes, discovers, measures, and represents. These actions are the core of the astronomers' work, and they do not change over time.

However, the historical instruments are still challenging to understand for visitors. To overcome this problem, the MARSS (MusAB in Augmented Reality from Science to Society) project aims at designing and developing a new digital journey inside the museum to allow different categories of visitors to enjoy the exhibition in an engaging and interactive way. The project's final goal is to create a cultural proposal to highlight how Osservatorio's research, science, and technology can offer surprising and unexpected positive social, cultural, and technological impacts. In particular, the project aims to innovate the cultural proposal of MusAB and give new life to the historical collection, illustrating the role that the Osservatorio has played from its foundation to today, with particular emphasis on the evolution of astronomy, technology, and culture. The MARSS project has been funded by the Fondazione Cariplo, started in September 2020 and lasts 30 months. The MARSS project is carried out at the Museo Astronomico di Brera (MusAB) by the project leader INAF–Osservatorio Astronomico di Brera, in collaboration with the Mechanical Engineering Department of Politecnico di Milano.

A specifically developed AR interactive application will illustrate the MusAB collection in an in-depth, comprehensive, and easy-to-understand way. The new interactive experience will be designed and created to "accompany" visitors within the museum collection through narrative paths and make obvious logical connecting elements that are not immediately deducible. Therefore, by using a "digital storytelling" approach, the meaning of the instruments, their using modalities, and their historical context will be illustrated.

This goal will be achieved through an approach that integrates virtual content, images, and audio content to allow each instrument to be framed in the appropriate path and to describe its characteristics. This will be achieved by using a gamification strategy to translate the content of high scientific value into more easily understandable elements. In this sense, the possibility of visualising the concepts instead of reading written explanations reduces the need for visitors with a scientific background and with very strong mental visualisation skills. It increases the possibilities of learning and memorisation. Additionally, the integration of different narrative modes and media involved, defined as "Transmedia storytelling" [35], will make the experience more engaging and complete for the visitors. Each medium, conveying complementary information, will contribute to the narration of the paths, asking the user to play an active role in integrating stimuli from the various media.

So, the project's primary goal is to create a "talking" museum collection, which can guide the visitor in the discovery of astronomy, its history, and the role of the Osservatorio in the context of the city of Milan. Furthermore, new interaction modalities will allow the visitor to interact with the provided digital content, creating greater intellectual and emotional involvement, and supporting the collection's self-discovery. In fact, the AR application has been designed as a storytelling interactive journey, which the users could experience autonomously. This goal has been identified to offer an innovative and beneficial alternative to the guided visits that the museum is currently providing. Moreover, due to the highly innovative features of this project, the MusAB AR application could be a valuable resource to attract a wider and younger audience to the museum and stimulate the current audience to consider future and more frequent visits. In fact, the interactive educational approach will "extend" the target audience to "non-expert visitors" who will better understand concepts often considered very complex through the new communication methods. In this way, even the non-expert visitors, fascinated by the idea of astronomy as a general theme but "scared" by the complex theories characterising the topic, will be able to approach them. School-age children can also be included in this category of non-expert

visitors, often very fascinated by observation of the sky but little aware of the type of study and instruments needed, in the past and today, to understand it.

*The Structure of the MusAB AR Experience*

The MusAB exhibition is mainly composed of historical instruments exposed inside the gallery and is organised into five main sections (see Figure 2). Graphic panels and captions support the communication of every section and briefly inform about the instruments. The first section introduces a question (*What does the astronomer do*?), to which the other sections try to answer. *Observe*, *Discover*, *Measure*, and *Represent* indicate the names of the remaining four sections, recalling the operations carried out by astronomers.

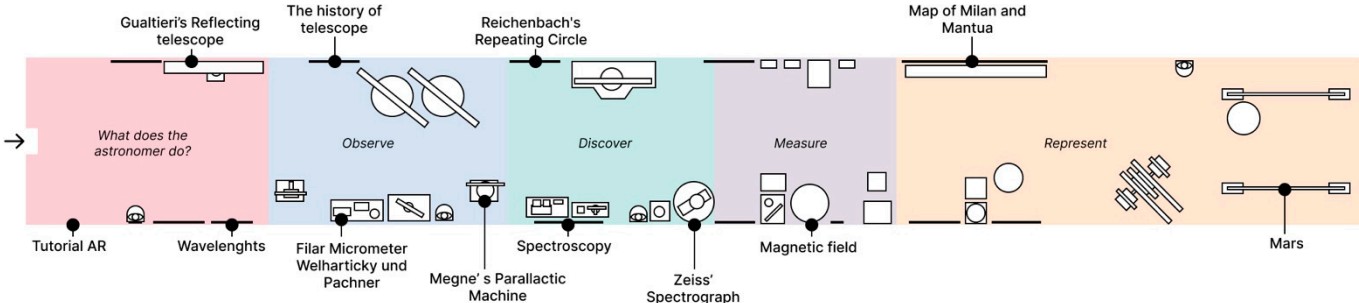

**Figure 2.** Scheme of the Gallery of Instruments with the division into sections and AR clips.

Similarly, the MusAB AR experience will be designed as a narration that develops along the five thematic paths, creating a link between history, science, and astronomical research. This narration will tell the story of the historical instruments displayed in the museum exhibition using AR technology. The advantage offered by AR will be to provide modular visits based on the type of public, meeting the interests of more experienced or less experienced audience and children. AR will also make the visit more interactive and engaging by conveying new content, starting from the instruments in the gallery that will be highlighted and enhanced. In addition, AR will allow visitors to explore the use of each scientific instrument at different levels of meaning and sense: from historical, social, and modern points of view.

The AR experience in the Gallery of Instruments is composed of different small augmented reality applications, named AR clips, that the user can enjoy by means of a tablet provided by the museum or by using his/her personal smartphone. A specific AR clip can be enjoyed for each selected historical instrument. Concerning the narration, the AR clips are not thought to be experienced sequentially. On the contrary, each clip is detached from the others by presenting self-standing contents. In this way, the visitor is free to explore the AR contents during the visit when she/he prefers, without being forced to follow a consecutive modality. Despite this free exploration, to facilitate the user experience, the clips have also been enumerated. Moreover, each AR clip presents a duration of approximately three minutes.

Two different narrative paths, one dedicated to adults and the other one for children (under twelve years old), have been defined based on the visitors target of the museum. These two paths are thought to be experienced both by parents and children individually or by parents and children together. The contents presented in the AR clips within the two paths are designed according to the target audience. In particular, the children's path is thought to be more interactive and less complex in presenting scientific theories, using less specific terminology and an adequate tone of voice. In addition, both the experiences were realised by integrating different sensory modalities. Indeed, each AR clip has visual and audio contents, presented simultaneously to the visitors. Moreover, both the adults' itinerary and the children's one can be experienced in Italian or English, allowing the user to switch between the two languages.

Specifically, in this paper, the authors mainly refer to adult clips, which have been designed and developed in the first 18 months of the MARSS project.

To support the experience, a mobile application was designed. The application will be freely available both on Google Play and on App Store and will be compatible with most of the commonly used Android and iOS devices, which can support augmented reality. This choice is dictated by the intention of facilitating access to the experience to a wide audience. Moreover, to guarantee the fruition of the experience also to people that do not have a personal device or that do not intend to download the application on their smartphones, the MusAB will provide some tablets that the visitors can borrow during the visit. The application proposes the division into sections presented in the gallery to facilitate the narration and the identification of contents by the visitors. Each AR clip can be activated by framing stickers (used as markers to activate the AR contents) located inside the gallery and linked to historical instruments. These stickers present the same name, number, and graphics of the mobile application contents. Most of the time, the names used to indicate the AR clips inside the application correspond to the name of the astronomical instruments' captions and panels located inside the gallery (reported in Figure 2). This aims at helping the user identify the stickers both inside the gallery and on the application.

When the user decides to start the AR experience, he/she can use the provided tablet or the smartphone to open the application. Then, he/she selects the clip that he/she wants to experience on the application, and the device camera is automatically activated. Consequently, the user frames the sticker corresponding to the selected clip to display the AR contents. Finally, once the single AR clip has been completed, he/she can come back to the main section page of the application to select another clip concerning the same section or to the main home page to choose another section.

As regards the digital content, it was specifically defined according to the objectives of each clip. As previously mentioned, the AR experience was designed and developed with different goals. One of the primary purposes was to allow the visitor to understand the history and functioning of the instruments displayed in the gallery to create an "active dialogue" with these objects. Indeed, the instruments were perceived as 'silent' artefacts, and most visitors could not understand their use, meaning, and story. For this purpose, the instruments have been analysed in detail to understand their functioning and to frame internal components and mechanisms. This gave the possibility to propose different types of visualisations in AR, creating 3D models, 3D exploded views and 2D section views, as for the clips named *Filar Micrometer Welharticky und Pachner* and *Gualtieri's Reflecting telescope*, showing to the visitors hidden parts of the instruments or recreating the original movements.

Further aims were to connect ancient astronomical instruments to the ones used nowadays, create a link between the past and the present, and present important discoveries. In response to these aims, ancient instruments were used as a medium for communicating scientific and astronomical concepts. Here, 3D models, 2D images and illustrations, and animations were used to narrate the augmented contents. All these elements were designed to support digital storytelling, proposing an intuitive explanation for users with no or little knowledge in the astronomical field. Indeed, the proposed contents, related to the adults' path, were defined for visitors with a medium degree of knowledge in astronomy, with some insights for expert visitors.

It is worth reporting that, for each clip, interactive elements were introduced to call visitors to the action, increase their involvement in the experience and improve their learning performance.

## 4. The Design Process

The design process of the AR MusAB experience is carried out by a multidisciplinary team composed of different members. The team comprises the authors of this paper, consisting of INAF astrophysics and science communicators, experts in virtual prototyping methods and technologies, plus external professionals such as a copywriter (Zelda Theatre

Company, https://www.zeldateatro.com/en/, accessed on 22 June 2022) for the storytelling and a tech provider studio.

For the development of the AR clips, the multidisciplinary team used a design iterative process composed of four main steps, *Research*, *Design*, *Develop* and *Test*, as represented in Figure 3.

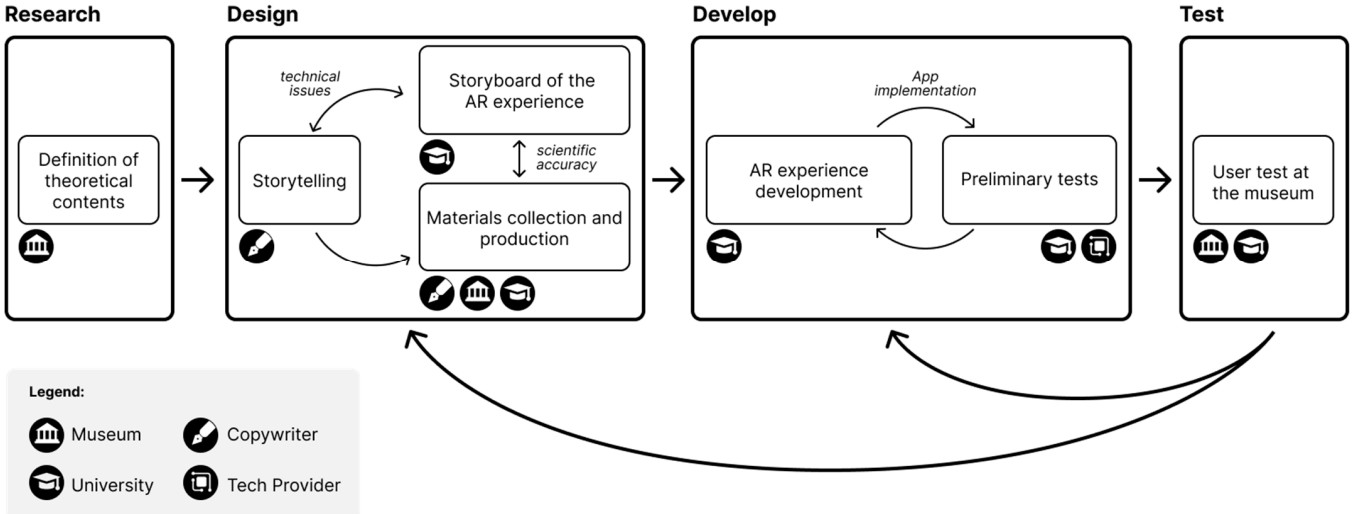

**Figure 3.** The design process.

The *Research* step concerned the definition of scientific and astronomical contents to be communicated for each historical instrument. This step had the purpose of selecting the most suitable astronomical contents to convey during the visit, and it has been mainly carried out by the INAF team. In the *Design* step, the contents were organised, applying the storytelling, written by the screenwriter of the theatre company, to create a multimedia narration. This regarded the creation of a story to connect all the previously defined elements and items of information to convey through the experience. Different choices were made considering the final user target. For instance, the tone of voice and the terminology level were set according to the target audience, i.e., adults with medium knowledge of astronomical concepts. In this phase, the involvement of a copywriter was crucial for the narrative creation. Then, starting from this structure, a storyboard for each AR clip was created. Specifically, the selected contents were analysed by experts in virtual prototyping, who then created the storyboard of the AR experience. Here, the different representation modalities and media were defined: 3D models, 2D illustrations, static or animated contents, and interactive elements. As the last step, the storyboard was evaluated from a scientific perspective by the INAF team to guarantee the correct transposition of theories and concepts and the compliance with the educational purpose of the experience. In the meantime, the material needed to create the AR experience was collected and produced. This material covered the selection of original images and photographs, the creation of illustrations, animations and audio tracks, and the collection and development of 3D models.

In some cases, the instruments exhibited inside the gallery were faithfully reproduced as 3D models. Other 3D models were instead downloaded from validated web sources, such as the NASA website, to ensure scientific accuracy. Moreover, audio tracks integrated to support the AR contents and enhance the multisensory stimulation were also recorded in this process phase.

During the *Develop* phase, the collected materials were put together, following the storyboard previously defined, for the development of the AR clip. During this phase, the experience was tested several times at the museum to evaluate and improve some aspects related to technical, usability, and communication points of view. Moreover, the visitor position during the fruition of the experience was tested, considering the location of the

exposed historical instruments and the possible flow of visitors inside the Gallery during the experience. At the end of this phase, the INAF team scientifically validated the developed AR experience, checking the correctness of the terminology and the representation of astronomical phenomena. After making any necessary changes, the AR clip was integrated into the specifically developed mobile application with the help of a tech provider studio. Finally, in the *Test* step, end-users were asked to test the AR clips inside the museum, and the analysis of the collected results highlighted the most appreciated aspects and the pain points of the AR clips. Specifically, the *Test* phase and the analysis of the obtained results are extensively described in Sections 6–8 of this paper.

### 4.1. Design the AR Content

During the *Design* phase, all AR clips were designed following a common structure, defined based on scientific and astronomical contents and technical requirements. Each clip was structured in three main parts: the first concerning the historical and scientific context connected to the instrument, the second presenting curiosity or particular facts related to the instrument or the context, and the last proposing the user interactions with the AR content. These three parts were proposed in each clip; the user is guided during the experience to complete all the steps. This structure represents one of the most valuable and challenging features of this project since it combines carefully selected scientific content, engaging storytelling modalities and gamified interactions to optimise the learning process and boost the visitors' engagement.

Considering the objects present inside the Gallery of Instruments and the aims to convey, different approaches were adopted to design the AR contents of each clip. Three main approaches can be identified:

- Virtual recreation of ancient astronomical instruments
- Augmenting the explanatory graphical panels
- Augmenting the ancient astronomical instruments

These three approaches were adopted considering the connection between the physical instruments and the topics addressed during the AR experience.

#### 4.1.1. Virtual Recreation of Astronomical Instruments

This first approach, used in several cases, consists of developing the 3D models of astronomical instruments to be used as one of the AR contents. This choice was also led by the fact that most instruments have huge dimensions, and some portions are hidden or not visible to the visitors. In addition, since the visitor is not allowed to touch the exposed instruments or manipulate them, the instrument's functioning and mechanism can hardly be understood. Instead, 3D models can be directly explored by the visitor from different points of view and by using different scales. Furthermore, by using these 3D models, animations were developed to simulate the instrument's real use and to show some hidden mechanisms and parts that can be otherwise complex to understand. This provides the visitors with the unique possibility to learn about astronomy by observing and interacting with ancient historical instruments, which otherwise would be perceived as extremely cryptic. For example, this approach was applied in the design of the AR clips concerning *Gualtieri*'s *Reflecting telescope* and the *Filar Micrometer Welharticky und Pachner*.

*Gualtieri*'s *Reflecting telescope* is a Newtonian telescope that uses mirrors to reflect light and display celestial bodies. In this case, the 3D model of the historical instrument was recreated for two main reasons: the instrument is huge and challenging to explore in all the details, and the internal parts are not visible. Hence, its functioning is difficult to understand. Instead, in AR, the 3D model can be inspected by the visitor through a section view used to reveal two mirrors inside the telescope tube. Moreover, the 3D model of the instrument was used as a starting point to address reflecting telescopes in general, their possible configurations presenting different numbers of mirrors, and to create a link with modern reflecting telescopes, showing pictures of the *Hubble Space Telescope* and *Telescopio Nazionale Galileo*.

*Filar Micrometer Welharticky und Pachner* is an instrument that is usually attached to the telescope for measuring angles and distances between celestial bodies. This specific instrument is stored in the museum's archive and is not exposed inside the gallery. However, in the AR experience, the virtual recreation of the instrument is displayed next to other Filar Micrometers presented inside a glass cage in the *Observe* section of the museum. During the AR experience, the visitor can understand the Micrometer functioning by closely watching all its parts. In addition, an exploded view was used to show the connection between the components, such as the knobs and the chassis of the instrument, as shown in Figure 4.

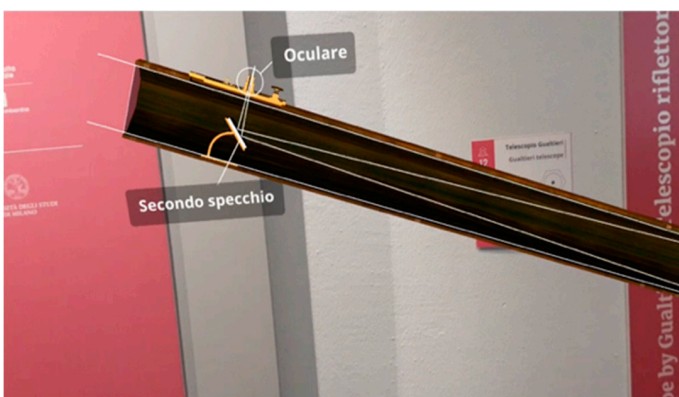 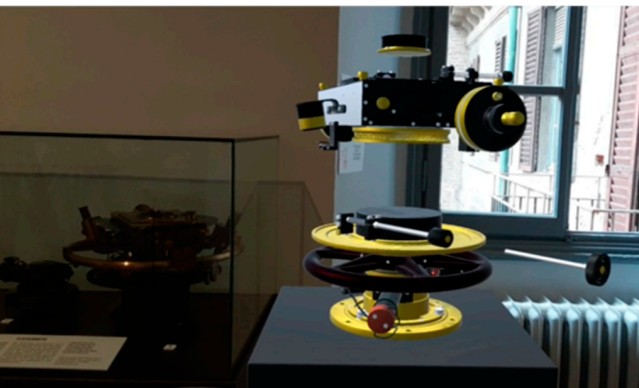

**Figure 4.** The Gualtieri's Reflecting telescope (section view, (**left**)) and the Micrometer Welharticky und Pachner (exploded view, (**right**)).

### 4.1.2. Augmenting the Explanatory Graphical Panels

The second approach concerned the use of explanatory graphic panels of the museum to present some complex concepts by providing augmented texts, images, and illustrations. Using AR content to "augment" these panels, the visitor can better visualise and understand these difficult scientific concepts by directly interacting with some elements. Moreover, the visitors can playfully interact with augmented contents extracted from the panels, which otherwise would often be ignored and considered less attractive compared to other assets of the museum.

This second approach was applied to the design of the *Wavelengths* and the *Spectroscopy* clips. In the *Spectroscopy* graphic panel several elements, such as the periodic table of elements and the star Sirius and its spectrum, are represented. These illustrations were used as a starting point for the narration created in AR. The *Wavelengths* graphic panel presents photography and illustrations of different telescopes that were used as a support for the AR contents. These telescopes can frame elements by using different wavelengths, showing different aspects of the same celestial body. In this case, the AR content allows a better understanding of the relation between the telescopes and the corresponding wavelengths, as shown in Figure 5.

### 4.1.3. Augmenting the Astronomical Instruments

The third approach used in designing a few AR clips consists of augmenting the astronomical instruments present in the gallery. This approach was adopted for the design of the *Spectrograph* clip. In fact, in this case, the instrument does not require the creation of a 3D model to be disassembled, showing all its components during the AR experience. In addition, this instrument is not located close to a wall, so the visitor can watch it from different perspectives just by walking inside the gallery. The same approach was also used to augment an ancient map, the *Map of Milan and Mantua*. As in the previous instance, it was not necessary to recreate the map virtually since the physical one offers good vertical support for presenting the AR contents on it. This approach offers visitors an exciting perspective to discover the collection of instruments of the museum. In fact, especially

when addressing a young audience, the AR technology gives the possibility to reveal and emphasise a surprising side of educational experiences.

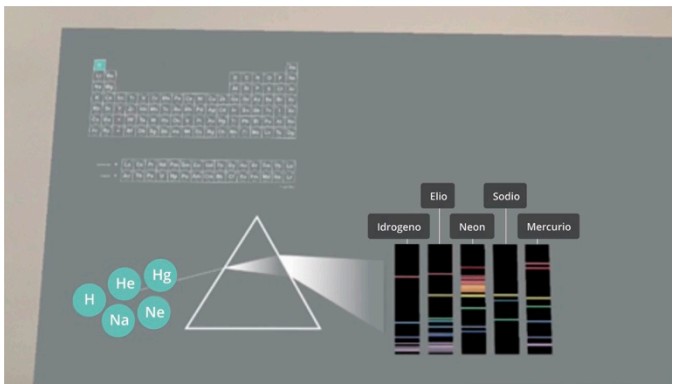
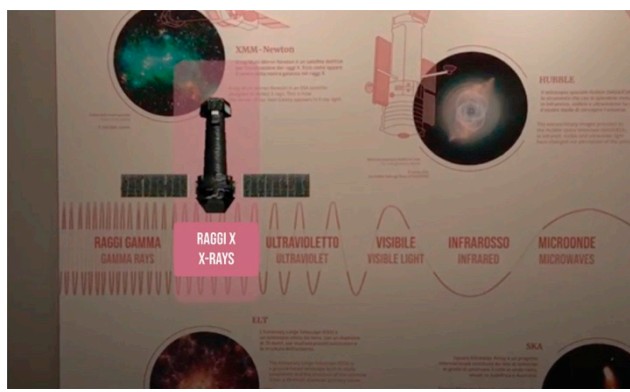

**Figure 5.** Augmentation of the graphical panels related to spectroscopy (**left**) and wavelengths (**right**).

The *Spectrograph* is an instrument that separates incoming light deriving from a celestial body by its wavelength or frequency and gives, as result, a spectrum. The real instrument was used as a support to display a schematic representation of its functioning in AR. The dimension of this object allows the user to frame it completely using the camera of the device easily. The most important features related to the instrument regard the position of internal mirrors that reflect the light to propose the spectrum of the celestial body (see Figure 6), which were digitally represented using bidimensional images, illustrations, and animations.

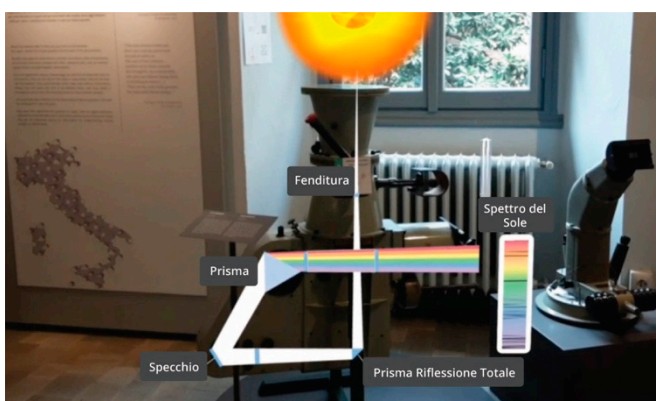
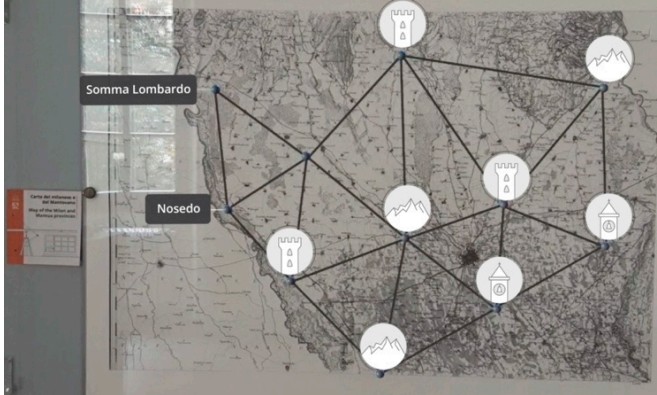

**Figure 6.** Augmentation of the spectrograph (**left**) and the Map of Milan and Mantua (**right**).

Using a similar approach, the *Map of Milan and Mantua* was used to display AR contents directly on top of it. The topic of the related clip regards the works conducted by astronomers in mapping the distances of physical places to create geographical maps such as the one exhibited. By using animations of 2D and 3D elements in AR, it was possible to show the visitor both the method and tools (geodetic rods) used by astronomers to map distances and realise the map.

### 4.2. Design the User Experience

As previously mentioned, some stickers placed inside the gallery indicate to the visitor the places in which he/she can enjoy the AR clips. By framing a sticker, the corresponding AR clip will start automatically. During the clip, the digital content is presented in a predetermined order (historical and scientific context, curiosity or particular facts related to the instrument or the context, interactive part), but the user can decide the timing of the fruition by using a *Stop/Play* button. At the end of each clip, the user can go back to

the application section to which the clip belongs. Moreover, a bar at the top of the screen presents a back icon to allow the exit of the clip whenever he/she prefers.

Typologies of User Interactions

As formerly introduced, an interactive part was integrated into each AR clip to engage the visitor and stimulate his/her active participation. These interactions usually occur at the end of the clip, after the more explanatory parts. For example, they allow the visitors to act on the virtual instrument to try its functioning.

Different interactive elements were integrated in each clip, both 3D and 2D. Some can be experienced in AR, while others were moved directly on the screen. This choice is related to complex interactions, in which visitors usually feel more comfortable interacting with touch screen elements (e.g., smartphones and tablets) than with augmented elements [36,37]. For the same reason, in some cases, graphical user interface (GUI) elements were used instead of interactions to accomplish with 3D models to simplify the visitors' actions. For example, in *Reichenbach's Repeating Circle* clip, the user can rotate the two axes of the instrument to catch a star orbiting around it by using two interface sliders, as shown in Figure 7.

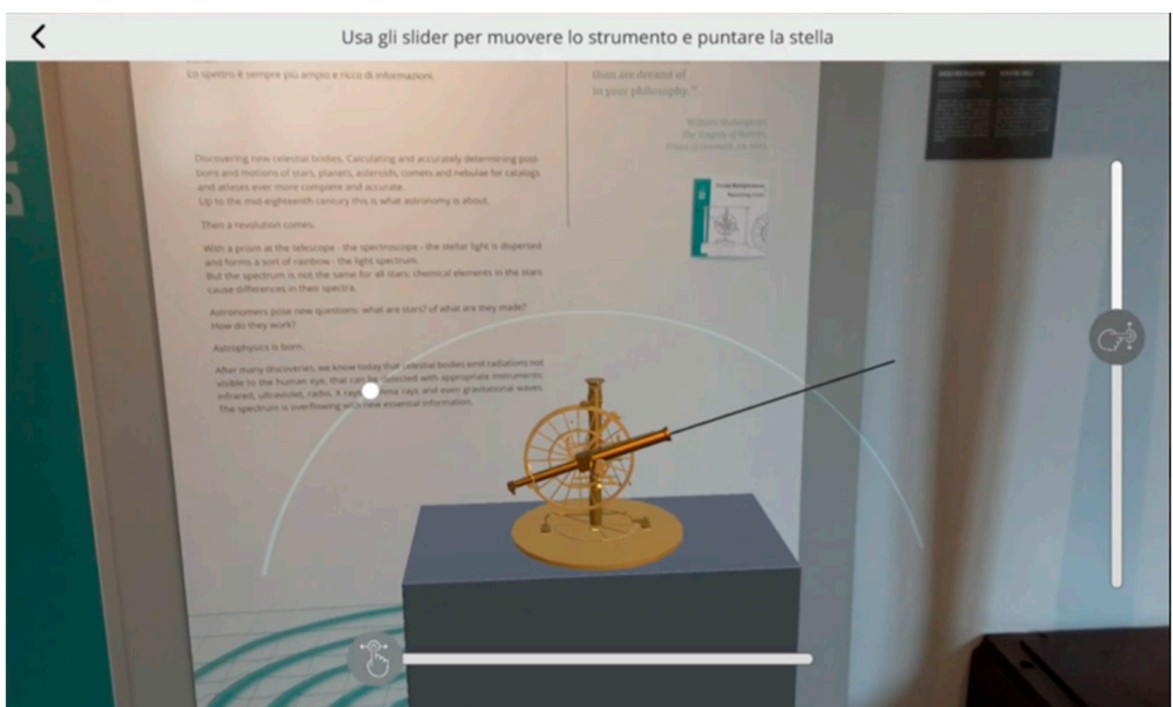

**Figure 7.** Reichenbach's Repeating Circle axes interaction through GUI elements.

Other methods were also adopted to simplify other direct complex interactions with 3D models. In the case of *Filar Micrometer Welharticky und Pachner*, the proposed interaction regarded the rotation of three knobs to a certain degree to move the instrument's chassis and its threads. In this case, an automatic movement was developed by using animation to help the visitor during the interaction and, at the same time, to give him/her the perception of influencing the rotation of the knobs.

Despite the main interactions, during the experience, visitors can also freely explore the virtual recreation of instruments, manipulating the 3D models and focusing on some aspects that otherwise will remain undiscovered. During these interactions, the visitor can decide the amount of time to dedicate to the exploration, having the possibility to manipulate and visualise the object in all the details and, consequently, decide to proceed to enjoy the rest of the clip.

The authors used different approaches to prepare the visitor for all the interactive parts and avoid frustration. Before entering the Gallery of Instruments, a *Tutorial AR* clip is introduced to provide a guide to the main interactive elements that the visitor will encounter during the AR experience. In this stage, an explanation regarding the most common interactions is proposed, and then the visitor is invited to interact with some elements to get familiar with them directly. Moreover, in each AR clip, a further tutorial presenting a video that shows the punctual interaction modalities is proposed just before the main interactive part. In addition, during the interaction, the visitor receives some indications both by audio and texts presented on the top of the screen, related to the actions performed in real time.

### 4.3. Graphical User Interface

The graphical user interface (GUI) elements of the AR experience were designed to consider the adult target, shaping the main communication elements, such as icons and texts, in a clear, coherent way during the experience. Indeed, a user interface (UI) design system was created (see Figure 8).

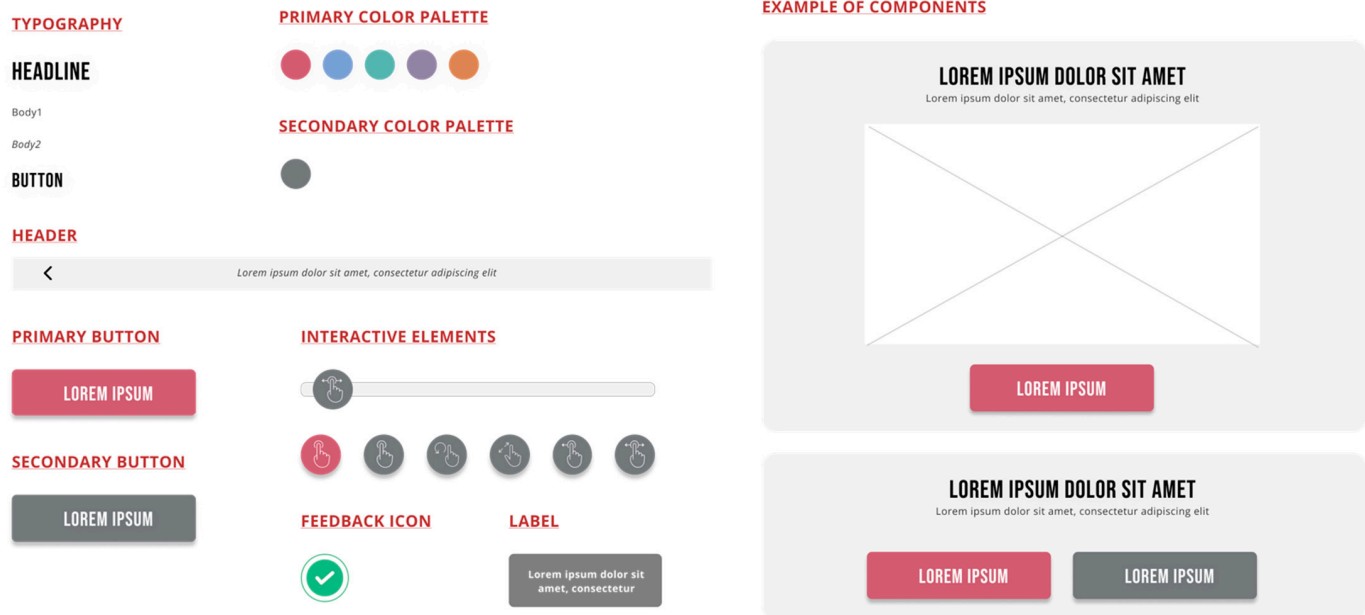

**Figure 8.** Main components of the user interface design system.

The UI design system was used in each AR clip to propose the same dimensions and spacing between the GUI elements and ensure visual coherence. Additionally, these elements were designed consistently in terms of fonts, shapes, and colours, starting from the museum brand identity, and recalling the graphic panels of the museum. The GUI elements are also related to the gallery section division, presenting the section's colour in which the AR clip should be experienced. The brand identity was taken into consideration also for creating the AR stickers to be attached inside the Gallery of Instruments. In addition, as previously mentioned, the texts and illustrations designed for the stickers were used in the application to guide the visitor in selecting the AR clips.

The GUI elements were designed by dividing them into two main categories: interface elements displayed on the screen and interface elements displayed in AR. For these two typologies, some different considerations were carried out. For the elements displayed on the screen, the most critical aspect considered their dimensions, which were carefully designed to avoid covering the AR contents. For example, some panels were displayed just during the experience. Regarding the interface elements displayed in AR, visibility and readability were mainly addressed, focusing on the written texts. In this case, to increase the contrast, a dark panel was placed as the background for the design of the AR labels. These

elements also follow the camera's points of view, in terms of rotation, to be consistently oriented toward the visitor.

Concerning the interface displayed in AR, another aspect was related to their disposition in the space. Indeed, the authors decided to adopt a horizontal interface configuration for the fruition of the AR clips. In this way, the user can better visualise all the AR elements, mostly presented with a horizontal layout. Despite this, some pages of the main application, such as the initial menu in which the interface elements are just displayed on the screen, were designed to be vertical (as in more traditional applications).

## 5. Development of the AR Clips

The *Development* part was made using mainly the Unity3D software (https://unity.com/, accessed on 22 June 2022). The 3D models were developed using Solidworks, Rhinoceros, and 3D Studio Max, while for 2D illustrations and animations, Adobe Illustrator and Adobe After Effects were used.

Concerning the project organisation, it was decided to create a unique Unity project in which each AR clip corresponds to a single scene. All the collected and developed materials have been integrated and used to create the AR interactive clips.

Different scripts were developed regarding the main interactive parts of the AR clips. Several 2D interactions present structures that are like overlapping puzzles or mazes games. Concerning the 3D interactions, scripts were developed to simulate the real functioning of some of the instruments of the gallery. Specifically, some dedicated scripts were developed to rotate some parts of the instruments by respecting a specific rotation range.

Some other complex interactions, usually concerning 3D objects, were achieved by using some Unity assets. The *Lean Touch* asset, which supports the rotation, translation and scaling of 3D models, was integrated to allow the manipulation and exploration of the virtual instruments by using specific touch inputs.

The *Mars* Unity package (https://unity.com/products/unity-mars, accessed on 22 June 2022) was used to support the AR content development and easily create Image Marker-Based AR clips. A sticker to be placed inside the museum gallery was created for each clip. The visitors frame the stickers to activate the AR content corresponding to the markers used inside Unity. Their graphic elements, such as texts and illustrations, were specifically developed to increase their tracking and readability by the device camera and the AR application. This feature is particularly important for developing stable and still visualisation of the AR content over real objects, even if the user stops framing the marker with the device or changes his/her position in the space, thus losing the marker tracking.

For this reason, AR content stability was an essential element considered during the development phase. Some clips related to the augmentation of the graphic panels required a greater accuracy in stabilising the AR contents. Indeed, in these cases, the content should precisely overcome the real objects. More complex AR markers were developed to increase the tracking precision, which was obtained by combining the stickers and an image representing the panel part on which the AR content will be displayed. This technique was used, for example, for the *Wavelengths*, the *Spectroscopy*, and the *Map of Mantua and Milan* clips.

Once the AR clips were completely developed, they were integrated into the mobile application. The application was developed also considering its possible scalability and that, in the future, new AR clips will eventually be designed and developed by following the above-presented process. Indeed, an important aspect of the adopted process is related to the scientific accuracy of the content presented, which needs to be proposed and revised accurately by the INAF team to be aligned with the educational purpose of the experience. This consideration is mostly related to the complex astrophysical topics proposed during the experience and their connection with the physical ancient instruments present in the museum. For these reasons, the possibility for users to be directly involved in proposing and creating contents, in an open platform perspective, was considered and then rejected.

As already mentioned, several testing sessions were carried out during the development phase to evaluate the technical performances of the AR clips. Consequently, extended testing sessions with users were performed and will be described in detail in the following sections.

## 6. Testing Session

In the *Test* phase, several testing sessions were performed to evaluate both the AR application's effectiveness in transmitting knowledge to the visitor and its usability. These aspects were chosen considering the main goals of the MARSS project. The main innovations of the cultural proposal of MusAB consist of the possibility to give new life to the historical collection and of the possibility to intercept the interest of a wider and younger audience of visitors. Therefore, the testing sessions have focused on evaluating the user experience of the AR interactive application and transmitting valuable knowledge to the participants. In addition, a subsequent goal was the validation of the AR clips, the identification of possible pain points and, at the same time, the most successful and effective elements in communicating information to the visitor.

Twenty-three voluntary participants were involved in the testing sessions at the museum. To avoid possible influence among other testing sessions, they were asked to reach the museum at different moments and days. At the beginning of each session, participants were asked to fill out a preliminary form on their own to collect some valuable background information for the analysis of the collected data. Then, they were accompanied into the gallery by one of the authors, who helped the participants in case of need during the experience and collected further information about the application's usability through observation. Finally, the participants were asked to use the AR interactive application.

Even though the AR experience was designed to be also approached in a non-sequential way, since each AR clip presents self-standing contents, most of the subjects autonomously approached the experience following the increasing numerical number of the AR clips, indicated both by the stickers and by the UI of the application. Only one participant out of 23, in the case of one single clip, decided not to complete it before moving to the next one. Once the participants fully explored the application, they were asked to individually fill in a second questionnaire to evaluate the overall AR experience and a third questionnaire to evaluate their learning performance. To conclude the testing session, participants were asked to join a short debriefing interview, to collect further information about their perception of the experience. The overall testing activity lasted around one hour per participant.

*Participants and Questionnaires*

The information about participants' backgrounds, collected through the first preliminary questionnaire, is presented in Figure 9. Those data were collected with a first preliminary questionnaire given to participants before the experience. The questionnaire, provided via Google Forms, is composed of (1) a section of closed-answer questions about demographics; (2) a section about the subjects' familiarity with technology—specifically AR—composed of closed-answer questions or ratings from 1 to 7 [38]; and (3) a section about the subjects' familiarity with astronomical knowledge, composed of closed-answer questions and ratings from 1 to 7.

Twenty-three participants took part in the tests. Their ages were between 15 and 72, which was the audience target identified for the MusAB. They were equally distributed by gender (12 male, 11 female) and generally had a middle-high level education. Moreover, 2 participants declared to be studying or working in the astronomical field, with a high level of knowledge of astronomical concepts, while 3 participants had a background in the high-tech field.

Focusing on the participants' familiarity with technology—specifically with AR—22 participants out of 23 declared to use smartphones every day. Additionally, all of them regularly use a personal computer. Moreover, they informed us of their middle-high

knowledge about new technologies, and that most of them have heard of AR before participating to this test. Only a few participants already tried this technology, but several declared to be very curious about AR.

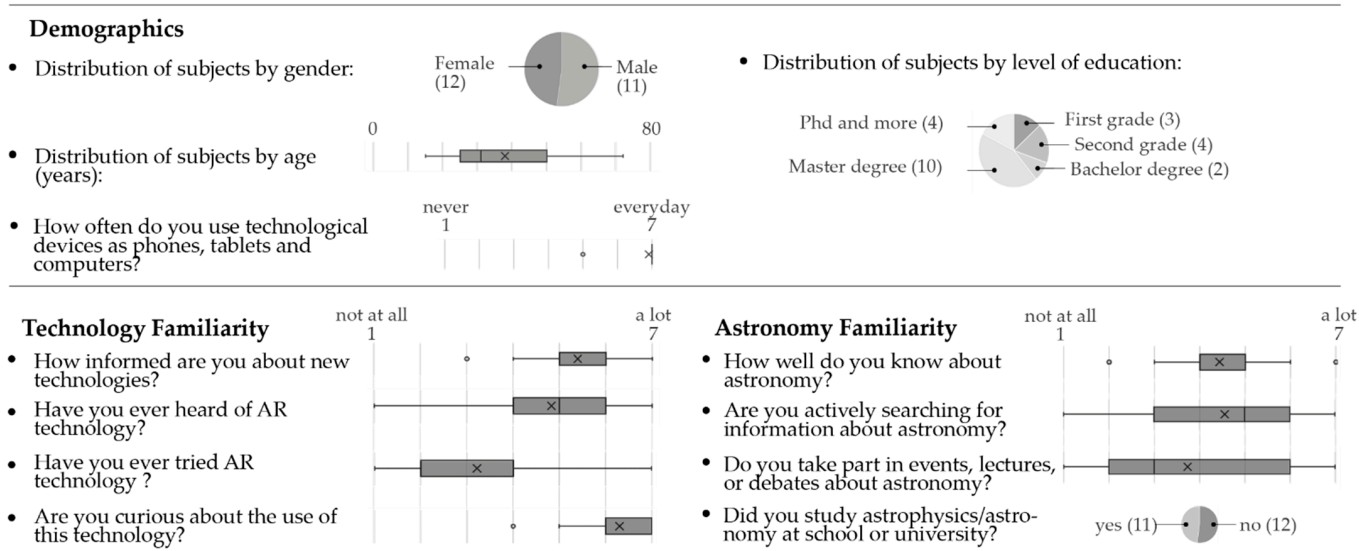

**Figure 9.** Participants' demographics and familiarity with AR technology and astronomy. The "X" symbol represents the average value.

Focusing on the subjects' familiarity with astronomical knowledge, they declared themselves as moderately informed about astronomy. Some stated to be active searchers of information about astronomy and regular followers of astronomical events, lectures, or debates. This approach in the selection of the participants is coherent with the identification of the MusAB target audience, characterised by a medium level of knowledge about astronomy, also taking into consideration expert visitors and newcomers.

The second questionnaire, based on [38–41], provided via Google Forms, focuses on participants' perception of the overall AR educational experience and its usability. Specifically, the first section is dedicated to participants' perception of the overall experience. They were asked to describe the experience with three adjectives, evaluate the overall experience and their engagement with a rating from 1 to 7, and declare if they have found any section frustrating. Moreover, participants were asked to evaluate the engagement and the difficulty of the interactive parts of the clips with a rating from 1 to 7, to indicate the best target audience for the experience by age, and to choose (through a rating from 1 to 7) if the experience could be better described as boring or engaging, relaxing or stressful, short or long, unpleasant or pleasant, useless or useful, alienating or engaging. A second section was used to evaluate participants' perception of the sense of presence and control on the overall experience, enjoyment, self-efficacy, quality of the information, ease of use, and usefulness.

Moreover, in this section, the participants' attitudes and behavioural intentions towards this and similar applications were evaluated. In this second section of the questionnaire, all the items were assessed with ratings from 1 to 7. Finally, participants were asked through open questions to communicate the aspects of the overall experience that were perceived as the most pleasant and memorable and if they would suggest changes in any element of the experience.

Lastly, the third questionnaire focuses on participants' learning performance. Specifically, participants were asked closed-answer questions with multiple choices on the educational content of each AR clip. Participants were asked three questions for each clip regarding (1) precise information about the instrument/its history (as an example, dates, specific names of astronomers, technical nomenclature, etc.); (2) information that could be derived by observing the visual AR content; and (3) technical information about the

instruments derived from interacting with the AR content, mainly focusing on "how the instrument works"

## 7. Analysis of the Collected Data

Focusing on the questionnaire evaluating the overall AR experience (Figure 10), the quality of the experience overall was estimated with an average rating of 5.6 out of 7, with a median value of 6. Additionally, the experience was evaluated as very engaging, with an average rating of 6.3 out of 7, with a median value of 7. Overall, the interactive sessions were considered very engaging and generally easy to perform.

The participants identified the best audience for the experience as adults and young adults (18 participants gave both answers). Additionally, the experience was found suitable for adolescents (15 replies) and preadolescents (13 responses).

The overall experience was described by participants as "engaging" rather than "boring", slightly more "relaxing" than "stressful", slightly more "long" than "short", "pleasant" rather than "unpleasant", and "useful" rather than "useless" and "engaging" rather than "alienating".

Both the visual and the audio stimuli of the experience were very engaging. Participants also have felt a good level of control over the events. They were able to predict the events caused by their actions on the AR content and focus on the activities required by the experience rather than on the functioning of the experience itself. Many participants also have declared to have learned through the experience new modalities of interaction that could improve their abilities with AR content. Therefore, we can state that the participants' sense of presence and control over the overall experience was positively evaluated.

Participants generally declared a high level of perceived enjoyment of the AR interactive application. They stated that the application was generally pleasant, regardless of its purpose. Participants have also communicated a high level of perceived self-efficacy while using the application by autonomously activating the AR content and using the application. The level of perceived quality of the information that has emerged from the questionnaire is high since participants considered the educational content of the application as trustable and clear. Moreover, participants generally considered the application a valuable source of information. Participants also communicated a high level of perceived ease of use of the application. They generally found the activity of framing the markers to activate the augmented content easy. They generally described the AR application as intuitive, easy to learn, and not requiring a great mental effort to approach it.

Generally, participants considered the application an experience of great value, which could offer a valuable point of view on astronomical education, help them understand the educational content, and stimulate the comprehension of astronomical topics. Regarding the participants' behavioural intentions, they would use the AR application again to get informed about astronomy. Additionally, all the participants appreciated the use of the AR application and consider it suitable for educational contexts. Moreover, participants agreed that a larger audience should have the opportunity to try the application. The application could also have the effect to push them to learn more about AR and to download and use similar applications in the future to learn more about astronomical topics. Participants would also be interested in returning to the museum to do the experience again when the MARSS project will be completed and in approaching similar experiences outside the museum's context. Lastly, participants would recommend using the application to others and would also use similar AR applications in the future to discover more about astronomy.

As shown in Figure 11, the adjectives most used to describe the application were "interesting", "interactive, "engaging", "entertaining", and "instructive". Therefore, it is possible to affirm that the AR application successfully conveyed positive emotions and values that could enrich the users' experience at the MusAB.

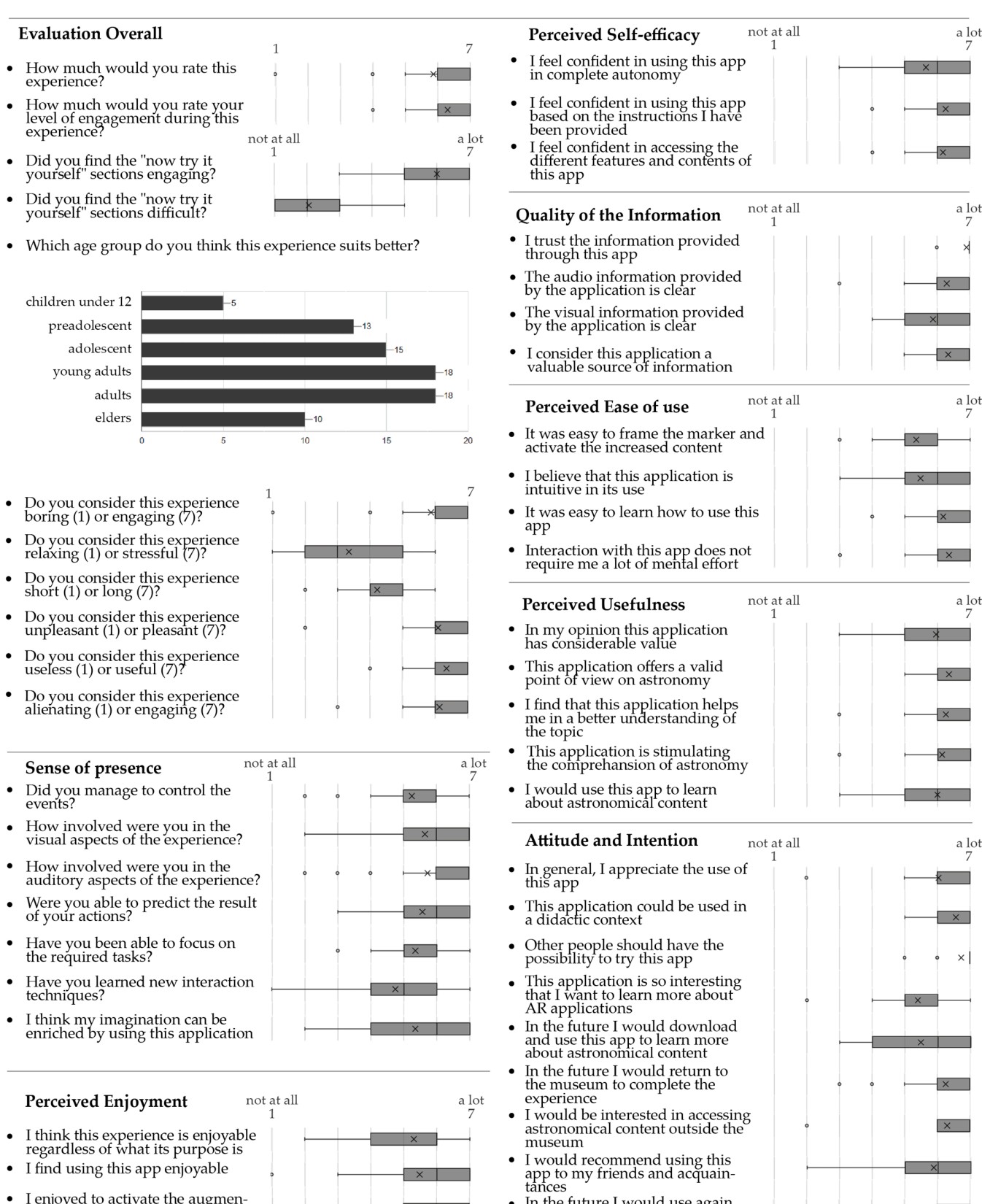

**Figure 10.** Data collected from the questionnaire evaluating the overall perception of the experience. The "X" symbol represents the average value.

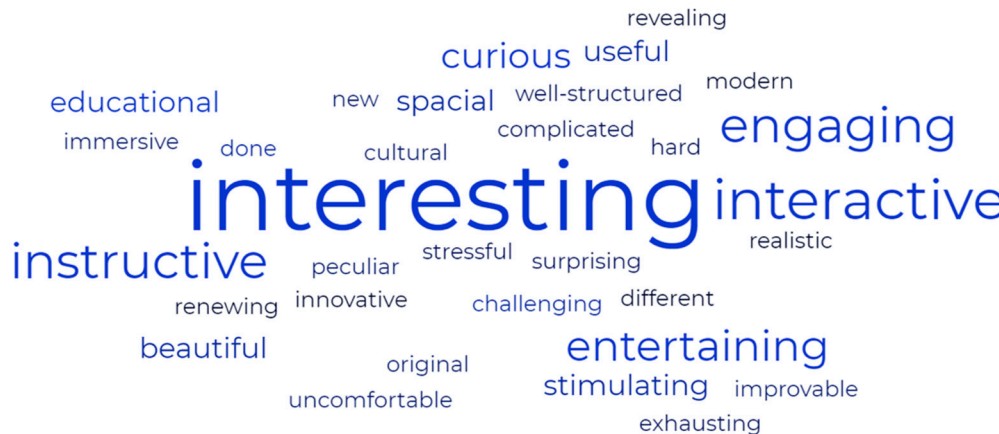

**Figure 11.** Word cloud to represent the adjectives used by participants to describe the application. The more often an adjective was used by the subjects, the larger the font of the corresponding word.

Focusing on the participants' learning performance (see Figure 12), most of the correct answers have occurred for scientific concepts enjoyed through AR representation, animations, and interactions. Instead, most of the answers given to the questions regarding specific data, such as technical nomenclature, names, and dates, were "I don't remember". Important exceptions occurred when these specific data were paired with a very peculiar notion or with an interactive section, as better explained later in the paper.

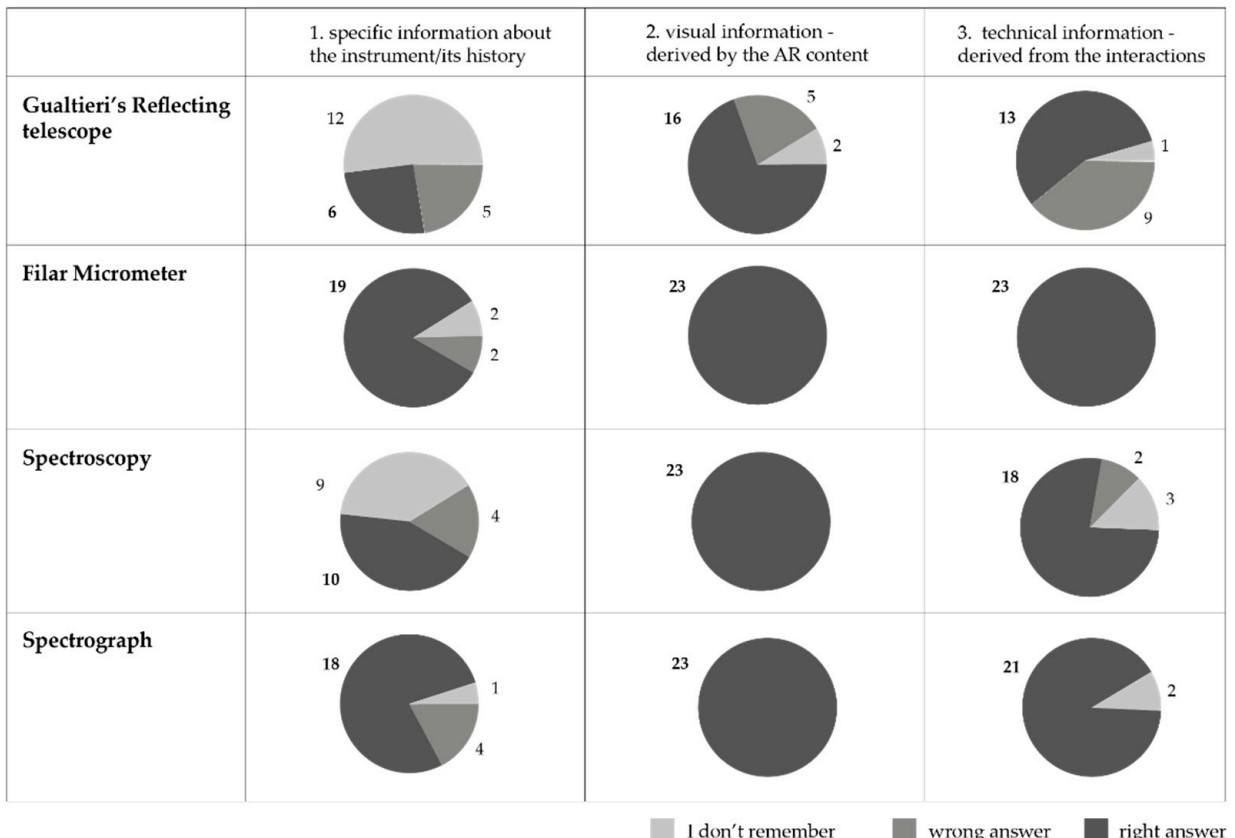

**Figure 12.** Distribution of the answers regarding the learning performance questionnaire. Specifically, the answers reported corresponding to the AR clips: Gualtieri's Reflecting telescope, Filar Micrometer Welharticky und Pachner, spectroscopy, and spectrograph.

## 8. Discussion

By observing the participants' behaviour inside the gallery while using the AR application and by analysing the collected data, it is possible to infer that the MARSS project is successful in satisfying the initial objective of translating the content of high scientific value into more engaging and easily understandable elements, autonomously accessible also for non-expert visitors. Additionally, observing the data collected through the questionnaires, it is possible to infer that the users have appreciated the AR experience and have reached a good level of learning preperformance, especially concerning the information provided through augmented reality and interactions with the digital content. After some indecision in approaching the first couple of AR clips, participants easily found the stickers positioned inside the gallery and were able to activate the augmented content provided by each of them autonomously. The interactions proposed to the users were generally evaluated as useful, engaging, and easy to perform, in accordance with [36,37]. The success of the interactions was also aided by the presence of both the preliminary *Tutorial AR* clip and the short tutorial presented before each interactive content of AR clip, which emerged as effective modalities to introduce the users to the interactive experience. Concerning the GUI designed for the AR experience, participants effectively understood it, have fluently moved inside the application, and understood how to proceed through the steps presented by each AR clip.

In the context of the MARSS project, it was not possible to perform a testing session with a control group. Nevertheless, some information extracted from the analysis of the data, both quantitative and qualitative, collected through the questionnaires and the final debrief with users allow us to positively evaluate the impact of the AR experience on the fruition of the content of the MusAB.

Concerning quantitative data, participants generally spent more time inside the Gallery than visitors who accessed it without any narrative support to facilitate their fruition of the content. In fact, the AR experience provides the users the possibility to have a narrating voice describing the content of the museum through an interactive storytelling modality, which the users evaluated as engaging and positively challenging. This narrative modality in communicating the educational content makes the visitors' stay at the museum more pleasant and interesting and, consequently, increases the duration of the visit.

Concerning qualitative data, participants described the integration of the AR experience inside the museum as a valuable and inspiring addition combined with the existing exhibit. In addition, from their point of view, the AR experience enhances the educational content of the MusAB.

Moreover, participants pointed out that the AR experience provides an innovative way to access the educational content of the MusAB through a narrative modality, which is only comparable to guided visits. However, guided visits are accessible only to a small number of participants at a time and require a greater effort from the museum staff. Therefore, the MARSS project can be considered a valuable resource, since the AR application is designed to be used also individually by the visitors. Furthermore, the participants of the testing session often communicated the intention of suggesting the experience to others and repeating the experience in the future, possibly enlarging the visitors' audience compared to the actual one, which is a key element in the final aims of this project.

Hereafter we will discuss a few of the AR clips tested during this experience to provide some exemplar cases useful to evaluate the pros and cons of applying AR to the relative educational content.

First, we consider the AR clips of *Gualtieri*'s *Reflecting telescope* and the *Filar Micrometer Welharticky und Pachner*. In these two cases, the astronomical instruments were virtually reproduced using 3D modelling to allow visitors to visualise all the instruments' components, rotating and scaling them and interact with them to understand their usage. Starting with *Gualtieri*'s *Reflecting telescope*, participants have appreciated the possibility of interacting with the object by manipulating the 3D model and observing its hidden components through AR. These elements have led to a good learning performance level

that has emerged from the collected data (see Figure 12). Specifically, participants clearly understood how the astronomers used the telescope, thanks to the 3D representation of its structure and a 2D simulation of the astronomer using the object itself.

Nevertheless, participants had difficulties understanding that the telescope is equipped with mirrors and not lenses. It is possible to assume that this aspect could be clearer if the actual interaction with those mirrors, given through a 2D interaction, would have been provided through the interaction with 3D models of these objects. Moreover, participants could not effectively grasp information regarding the historical age in which the telescope was acquired by the MusAB, probably because this information is not sustained by any specific AR content and is just explained by the audio description.

Instead, regarding the AR clip of *Filar Micrometer Welharticky und Pachner*, participants communicated a very high level of learning performance (Figure 12). They were able to successfully recall information about the usage of the instrument and its purpose, which were communicated through 3D representations and direct interaction with the augmented instrument. In this case, participants also remembered the specific names of the astronomers cited through the narration, probably because this information was supported by AR representation of the specific concept. Lastly, it is essential to take into consideration that, even though the interaction with the 3D content led to good learning performance, it was considered among the hardest to perform by the audience because of its realism, even if it was simplified during the design phase to facilitate the user interaction.

Another interesting case is the AR clip relative to *Spectroscopy*. In this case, a different approach to AR was taken to communicate the educational content. The panel's content was augmented since a physical instrument is absent in this case. The content of this section was generally considered engaging, and the learning performance derived from it can be considered good. Participants were able to grasp the most essential concepts communicated by the AR clip thanks to their augmented representation (see Figure 12).

Nevertheless, in this case, participants had difficulties remembering specific information connected to the names of relevant astronomers, probably because they were not fully associated with specific AR representations. Moreover, this AR clip was considered the most unpleasant by participants because of two reasons. Firstly, since the AR content needs to be perfectly overlapped with the physical panel present inside the museum, imperfections in the position of the augmented content are more visible and have an impact on the user's perception of the experience. Moreover, participants generally found the positioning of the AR content uncomfortable since it forced them to stand still with the device framing a high section of the panel for quite a long time, causing them to become tired by this action quickly, affecting their perception of the pleasantness of experience and their ability to concentrate on the educational content.

Lastly, the most appreciated AR clip has been the *Spectrograph*. In this specific clip, participants can directly superimpose the augmented content on the real instrument. This approach to AR was the most appreciated since it granted participants the ability to focus simultaneously on the real object and the augmented content. Their appreciation of the AR segment had positive consequences also on their learning performance (see Figure 12) since they were able to effectively grasp information on both the instrument functionalities, its purposes and specific historical details regarding the instrument itself.

## 9. Conclusions

The research work presented in this paper proposes to improve the user experience of a science museum using digital technologies. Specifically, the MARSS project was designed and developed to create a new digital journey inside the MusAB to improve users' engagement and understanding of the topic. Particular attention was devoted to the innovation of the cultural proposal of MusAB and giving new life to the historical collection.

As formerly introduced, an AR interactive application was specifically developed, and several testing sessions with users were performed. Based on the analysis of the collected data, it is possible to affirm that the MARSS project is successful in translating the content

of high scientific value into more engaging and easily understandable elements and that the users appreciated the AR experience and reached a good level of learning preperformance, especially concerning the information provided through augmented reality and interactions with the digital content.

However, based on the analysis of the collected data, some improvements will be carried out in the future. The first improvement concerns the visitor's position inside the gallery while experiencing the AR content. Because some participants found some difficulties navigating the gallery's overall content and identifying the correct distance to approach the augmented content, more attention will be paid to effectively highlighting, also through some recognisable wayfinding assets, the location of each AR clip in the gallery. Additionally, for some clips that were positioned relatively high or low concerning the users' point of view, improvements could be carried out to make the AR content more comfortable.

Finally, the fruition of the AR content of some sections of the experience could be improved considering the following aspects. The AR content shown on screen should never be too static to keep the viewer engaged during the overall experience. In addition, the congruence between the visual and audio stimuli provided by the application should be maximised to facilitate the learning process by reinforcing the communication of the information. Additionally, maintaining a constant structure composed of a theoretical introduction and an interactive part was evaluated by participants as a winning choice since it helps them to understand the interaction with the augmented content and to predict the consequences derived from it. The application structure could also be improved by balancing screen-based and AR-based parts to develop a more approachable and less challenging experience for visitors. Indeed, some participants argued that a long AR session, which requires them to keep the device up framing the marker, could be quite tiring and affect the overall perception of the experience and their attitude to learning.

In the subsequent phases of the MARSS project, these elements will be used to improve the already-developed clips and design the new AR clips dedicated to adults and children (under 12 years old). They will be further validated in the final testing sessions of the project.

**Author Contributions:** Conceptualisation, E.S., S.P., M.B., I.A., L.B. and M.C.; Methodology, I.A. and M.C.; Software, E.S. and S.P.; Supervision, I.A. and M.C.; Validation, E.S. and S.P.; Writing—original draft, E.S., S.P., M.B. and M.C.; Writing—review and editing, I.A. and L.B. All authors have read and agreed to the published version of the manuscript.

**Funding:** This research has been co-funded by Fondazione Cariplo, under the call "Luoghi di innovazione culturale—2019" (project: MARSS, MusAB in Augmented Reality from Science to Society, code 2019-3884).

**Institutional Review Board Statement:** Not applicable.

**Informed Consent Statement:** Informed consent was obtained from all subjects involved in the studies.

**Data Availability Statement:** Not applicable.

**Conflicts of Interest:** The authors declare no conflict of interest.

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
