# Peer review of "Augmented Reality to Engage Visitors of Science Museums through Interactive Experiences"

_heritage, doi:10.3390/heritage5030071_

Round 1
Reviewer 1 Report
Introduction and Related Works:
The rationale for increasing interactivity and storytelling is agreeable and the connection of these changes to the affordances of virtual and augmented reality is also fair, with examples used well to clarify the authors’ positions. Some of the examples in related works were difficult to envisage from the description given (e.g., the MONA “O” device). It would have been helpful to see explicit reference to evidence of positive impact directly attributable to these related works. This would more fully reinforce the value of such technological interventions with quantitative data that could justify the financial implications. Within this section I would also have been interested to read the authors’ perspectives on not just how the technology is used, but what the key features, qualities and affordances are that provide the added value to visitors. Are there specific design features, interactive mechanics, gamification techniques (etc.) that stand out?
The MARSS project and the Design Process:
The context of the use case is well defined and its challenges for visitors explained clearly. The underpinning theoretical perspective of transmedia storytelling is convincing and aligns with a lot of recent thinking on museum enrichment. The pitch-writer in me would suggest that these sections could champion the innovative features of the MARSS project over other use cases a bit more. The descriptions given are very detailed but I felt the unique qualities of MARSS could be further emphasised. Illustrations are used very well throughout and give a clear indication of what the MARSS system is and how it works from the user-perspective. Both separate and integrated narrative paths for adults and children is a really nice touch.
The design process is comprehensive and clear. All core aspects of AR design I feel have been accounted for and it was good to see interactive elements and GUI design considerations given a strong representation in the report. I feel that it would be helpful to include details on deployment as this can be a crucial (if boring) part of the user-experience. Would the system be accessible through separate app builds on the Apple Store and Google Play, or some kind of open platform/progressive web application accessed via the museum website?
Testing:
The sentence between 614-617 feels a bit awkward and I think has a typo on line 615. The setup is well-described and the demographics table is a nice way of efficiently communicating a lot of participant detail at a glance. The construction of the questionnaire I would say is fit for purpose with appropriate depth and good coverage of relevant data points. The acquisition of learning performance as well as subjective visitor-perception metrics is very much appreciated and the pre/post comparison is appropriate. The discussion considers future design changes at a good level of depth and the conclusions are arguably a fair reflection of the results. My main issue with the testing is in the apparent lack of a control. Although I will say that the results are impressively positive, many existing use cases report positive user-response and learning outcomes but lack a control for us to statistically interpret the added value of the technology against a ‘traditional’ museum installation. Whilst I don’t feel that the study needs repeating to justify publication of this work, I would ask if the authors could provide any further indication of how their AR intervention affected user-response and learning outcomes more than experiencing the Gallery without it.
Reviewer 2 Report
The originality of the proposed paper is reflected by the various developed augmented reality integrated within the MARSS project. The applications have been designed to enhance the users`s experience within the Museo Astronomico di Brera from Milan.
The paper is organized and well documented. Within the introduction, the authors have presented various aspects regarding the design and implementation of modern interactive museum exhibitions that integrate digital technologies. To facilitate and support both storytelling and immersive interaction, the most common used technologies are based on virtual reality and augmented reality technologies. The research aims and goals are presented at the end of the introduction along with the challenges associated with the development of an interactive digital exhibition that integrates AR technologies. The related section is well documented with a good number of recent references regarding the development of VR and AR experiences for museum exhibitions. The following section presents the Museo Astronomico di Brera collection along with the research project aims and goals. The structure of the MusAB AR experience is presented in detail and integrates a wide variety of applications that present various aspects such as wavelengths, spectroscopy, and magnetic fields. The following section is focused on the design process which involves a multidisciplinary team and a four-step design process (research, design, develop and test). The digital augmenting of various graphical panels and astronomical instruments along with the design of the user experience is presented within the following section. The following sections provides information regarding the testing sessions which made use of 23 participants as well as the analysis of the questionnaire evaluation of the collected data. The discussion and conclusions sections are based on the authors findings regarding the use of various AR applications integrated within the science museums.
The paper includes a small number of sources and research papers related to the development of innovative museum exhibitions that integrated digital technologies such as virtual reality and augmented reality. This section could be extended to integrate a higher amount of recent published papers (during the last five years) regarding the development of innovative museum exhibitions.
The proposed research paper has a high interest for readers and scientists that develop digital applications intended for innovative museum exhibitions as it presents a great case study example regarding the development of interactive museum experiences within the Museo Astronomico di Brera science museum.
The proposed research paper presents the implementation of the MARSS project aimed at designing and developing of a digital journey/experience inside the Museo Astronomico di Brera science museum based on special --tailored augmented reality storytelling application aimed to engage visitors.
-------
As specified before, my only concern with the proposed paper is regarding the small number of recent published research paper references. I would suggest at to add at least 3 more recent published papers aimed on the design, development and implementation of digital technologies (AR,VR, XR, natural gestures sensors) within museum exhibitions.
